# NEP: Autoregressive Image Editing via Next Editing Token Prediction

**Huimin Wu**[1]  **Xiaojian Ma**[1]  **Haozhe Zhao**[2]  **Yanpeng Zhao**[1]  **Qing Li**[1]✉

[1]State Key Laboratory of General Artificial Intelligence, BIGAI  [2]Peking University

**Project website**: nep-bigai.github.io

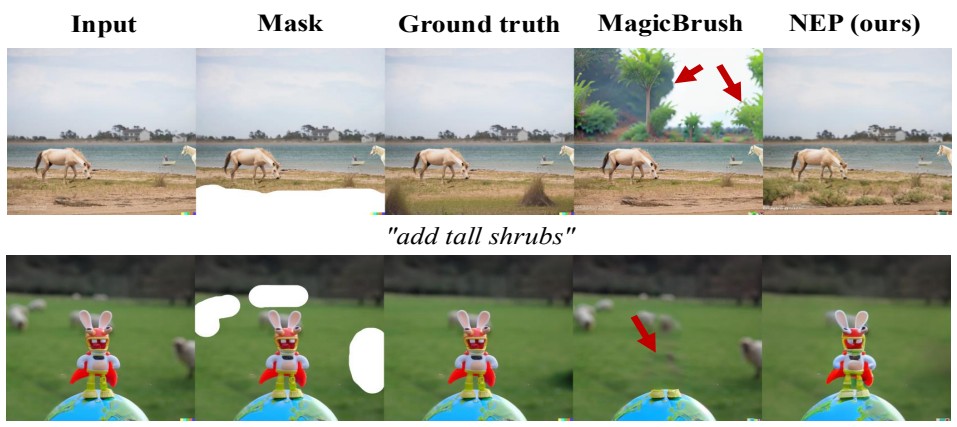

Figure 1: Our approach avoids full-image generation and does not introduce unintended changes as the previous diffusion-model-based editing approach [58].

## Abstract

Text-guided image editing involves modifying a source image based on a language instruction and, typically, requires changes to only small local regions. However, existing approaches generate the entire target image rather than selectively regenerate only the intended editing areas. This results in (1) unnecessary computational costs and (2) a bias toward reconstructing non-editing regions, which compromises the quality of the intended edits. To resolve these limitations, we propose to formulate image editing as **Next Editing-token Prediction** (NEP) based on autoregressive image generation, where only regions that need to be edited are regenerated, thus avoiding unintended modification to the non-editing areas. To enable any-region editing, we propose to pre-train an any-order autoregressive text-to-image (T2I) model. Once trained, it is capable of zero-shot image editing and can be easily adapted to NEP for image editing, which achieves a new state-of-the-art on widely used image editing benchmarks. Moreover, our model naturally supports test-time scaling (TTS) through iteratively refining its generation in a zero-shot manner.

## 1 Introduction

Text-driven image editing aims to modify a source image following a given language instruction. Typically, modifications are confined to small local regions (editing regions) while most of the image remains unchanged (non-editing regions). A predominant paradigm for solving the task is through

39th Conference on Neural Information Processing Systems (NeurIPS 2025).

the diffusion model [40, 39, 31], but standard diffusion models struggle with controllable editing, that is, editing only a target region without altering the surrounding areas. To tackle this challenge, an inversion technique has been proposed and augmented with diffusion-based image generation models [39, 21]. The core idea of this method is to find the mapping of non-editing regions to the corresponding subspace of Gaussian noise. It requires that the initial Gaussian noise that can be decoded into the source image should be pre-defined, which is, however, hard to obtain exactly, and further leads to unintended edits [11].

A more controllable paradigm is to pre-define editing regions and edit only the specified areas while preserving the rest [1, 23]. However, these approaches perform full generation of the target image, including regions that are not required to be edited, and thus are suboptimal in terms of efficiency. This inefficiency is pronounced in training-based editing approaches [2, 50], which also demand significant computational resources to learn to reconstruct. Moreover, the reliance on full-image generation introduces a learning bias during training; that is, image editing models tend to prioritize reconstruction for the non-editing regions over regeneration for the intended editing regions [50].

To address these issues, we introduce **N**ext **E**diting-token **P**rediction (**NEP**), a new formulation of text-guided image editing based on autoregressive (AR) image generation. NEP primarily focuses on regeneration for the editing region and removes the need for optimizing reconstruction for the non-editing areas. Consequently, it improves efficiency and circumvents the learning bias simultaneously. Since the standard AR model employs a fixed raster-scan generation order, it is incompatible with NEP's requirement to generate arbitrary editing regions. To address this, we develop NEP using a two-stage training strategy. First, we pre-train RLlamaGen, a robust random-order AR-based text-to-image (T2I) model that supports arbitrary-order generation and zero-shot local editing. In the second stage, we fine-tune RLlamaGen to optimize NEP's editing performance. Additionally, NEP enables test-time scaling through iterative refinement, improving generation outcomes. We summarize our contributions as follows:

- We propose a new formulation of image editing as next editing-token prediction. It simplifies the learning objectives to regeneration only, leading to higher efficiency and better editing quality. Our approach sets up new records on region-based editing tasks and achieves competitive results on free-form editing benchmarks.

- We propose a two-stage training regime for NEP, where the first stage creates RLlamaGen, a new T2I model capable of arbitrary-order full image generation and zero-shot local editing.

- We analyze the test-time scaling behaviors by embedding NEP in an iterative refinement loop.

## 2 Methods

In this section, we first introduce the pre-training approach RLlamaGen that can generate image tokens in any user-specified order (§2.1). Then, we elaborate on NEP for image editing (§2.2). Finally, we introduce test-time scaling strategies (§2.3) by integrating NEP in an iterative refinement loop.

### 2.1 NEP Pre-training

**Preliminaries on LlamaGen.** The NEP framework is versatile and compatible with various design choices[42, 49, 48]. In this work, we build upon LlamaGen[42], the first open-source text-conditioned autoregressive model to outperform diffusion models, leveraging its robust architecture to enable NEP's random-order generation and iterative refinement for enhanced image editing and generation. To maintain potential unification with text modality, the architecture design of LlamaGen largely follows one of the popular LLMs, Llama [45, 46]. The conditioning text embeddings are extracted from FLAN T5 [7], followed by a projector for dimensionality alignment. The text embeddings are left-padded to a fixed length $L_T$ and prefilled to generate image tokens. Images are firstly tokenized by the encoder and quantizer of VQGAN [8], and generated token ids are mapped to RGB pixels by the decoder. Image tokens with length $L$ are generated in a next-token prediction fashion. Formally, given a text sequence $T$, the sequentialized image tokens $I = \{I_1, I_2, ..., I_L\}$, are generated by:

$$p(I) = \prod_{i=1}^{L} p(I_i | I_{1,...,i-1}; T) \tag{1}$$

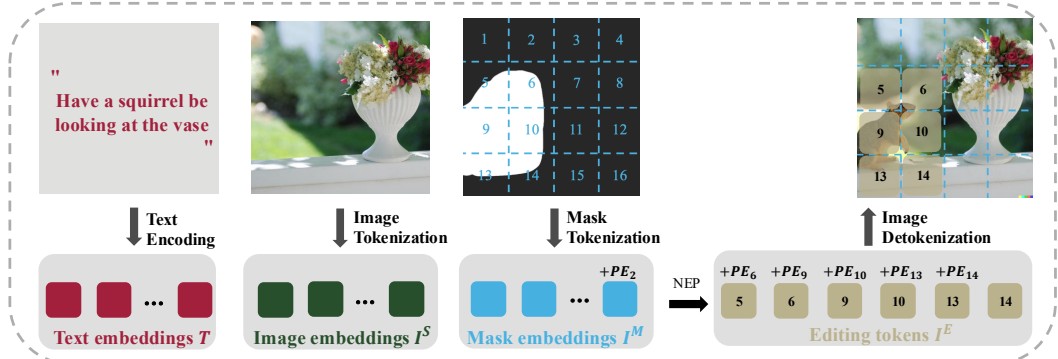

Figure 2: **Overview of Next-Editing-token Prediction.** The input sequence is comprised of: 1) **text embeddings**, extracted from FLAN-T5, 2) **source image embeddings**, tokenized by VQGAN, and 3) **mask embeddings**, a sequence of interleaved editing and non-editing embeddings. The output **editing tokens** (in raster scan order) are filled back to the source image based on the editing mask. $PE_i$ denotes the learned positional embeddings that specify the token generation order.

**RLlamaGen: Randomized Autoregressive Text-to-Image Generation.** To address LlamaGen's limitation of generating image tokens solely in raster scan order, we extend it to create RllamaGen, which supports generating image tokens in any user-specified order, enabling flexible, arbitrary-order generation [26, 55, 25]. To add order awareness to the model, following [26, 55], we learn an extra sequence of positional embeddings $PE_1, PE_2, ..., PE_L$, which is shuffled based on a random order to define the generation sequence. For each input image token, the positional embedding corresponding to the next token in the assigned order is added. Formally, the generation of an image sequence $I_O$ in the order of $O = [o_1, o_2, ...o_L]$ is defined as:

$$p(I_O) = \prod_{i=1}^{L} p(I_{o_i}|I_{o_1} + PE_{o_2}, ..., I_{o_{i-1}} + PE_{o_i}; T) \tag{2}$$

RLlamaGen supports zero-shot editing by regenerating tokens at given positions, allowing seamless transferability to image editing.

## 2.2 NEP: Next-Editing-token Prediction

NEP leverages three types of conditioning for region-based editing: 1) text instructions tokens, 2) source images tokens, and 3) editing region masks tokens. The tokenization of text instructions and images remains consistent with the pre-training stage. We detail the construction of editing region conditioning sequences derived from a pixel-level mask $M \in \{0, 1\}^{H \times W}$.

**Editing Region Conditioning** (ERC) We firstly patchify the pixel-level editing mask $M$ by max-pooling each non-overlapping sliding window with the size of $p \times p$. Subsequently, we flatten the patched mask into a sequence $M^E = \{m_1, m_2, ...m_L\} \in \{0, 1\}^L$. The masking sequence $I^M = \{I_1^M, ..., I_L^M\}$ is tokenized by querying a two-sized codebook comprising an editing embedding $E_{emb}$ and a non-editing embedding $U_{emb}$, which is formally defined as:

$$I_i^M = \begin{cases} E_{\text{emb}} & \text{if } M_i^E = 1 \\ U_{\text{emb}} & \text{otherwise} \end{cases} \tag{3}$$

Our editing model processes $L_T + 2 \times L$ input tokens and generates $L_E$ editing tokens, corresponding to the masked target image tokens, denoted as $I_E$. The generation order corresponds to the positions of the editing tokens within the raster scan order, denoted as $O^E = \{o_1^E, ..., o_{L_E}^E\}$.

Formally, our NEP strategy is defined as:

$$p(I_E) = \prod_{i=1}^{L_E} p(I_{o_i^E}|I_{o_1^E} + PE_{o_2^E}, ..., I_{o_{i-1}^E} + PE_{o_i^E}; T, I_{1,...,L}^S, I_{1,...,L}^M) \tag{4}$$

In scenarios where a region editing mask is unavailable or for global editing tasks (e.g., style transfer), the editing tokens are predicted according to the raster scan order.

## 2.3 Test-time Scaling with NEP

NEP can be employed to support test-time scaling by integrating it into a self-improving loop. In each refinement step, prior to NEP, a revision region is proposed. Existing image reward models [51] usually produce a single value for the full image. To obtain token-level dense quality scores, we calculate Grad-CAM [35] value regarding the critic model (*i.e.,* off-the-shelf CLIP-ViT-B/32). These values reflect each token's contribution to the overall image quality score, measured by a reward model (*i.e.,* ImageReward [51]). Positions that correspond to the $K$ lowest scores are identified as the revision regions. During revision, we adopt NEP to regenerate tokens in this region, conditioning them on the remaining high-quality tokens. After NEP, the reward model evaluates whether the revised image surpasses the original, determining whether to accept or reject the revision. To further improve quality, for NEP, we apply a rejection sampling strategy, regenerating tokens at the revision positions in multiple random orders and selecting the revision with the highest quality score. This approach demonstrates strong scaling potential, suggesting that effective revision of initial generations can significantly enhance performance.

## 3 Experiments

We evaluate our framework on the image editing and text-to-image generation tasks. Firstly, we introduce the full training setup that trains the RLlamaGen and NEP stage-by-stage (§3.1). Secondly, we evaluate NEP for image editing and validate its design choices from various aspects (§3.2). Then, we demonstrate the results of NEP pre-training model RLlamaGen (§3.3). Finally, we showcase the test-time scaling behaviors (§3.4).

### 3.1 Datasets and Training settings

**T2I pre-training settings.** We use LlamaGen-XL with 775M parameters as the base T2I model and adapt it to RLlamaGen adding 0.3M positional embedding parameters. Our training data consists of around 16M text-image pairs and is collected from multiple open-source datasets, including ALLaVA-LAION [5], CC12M [4], Kosmos-G [24], LAION-LVIS-220 [34], LAION-COCO-AESTHETIC [18], LAION-COCO-17M [56], and ShareGPT4V [6]. We train RLlamaGen for $60,000$ steps with a batch size of 360 and an image resolution of $256 \times 256$. The optimizer is Fused AdamW with $\beta_1$, $\beta_2$ set to 0.9, 0.95, respectively, and a constant learning rate of 1e-4 is used. We perform training on 8 NVIDIA Tesla A100 GPUs, which takes 39 hours.

**Image Editing Training Settings.** We fine-tune RLlamaGen for image editing by adding two learnable embeddings (i.e., $E_{emb}$ and $U_{emb}$) to specify masking regions. This strategy is computationally efficient, with only 3.6k parameters introduced. Our editing model is trained on the UltraEdit dataset [60] that comprises 4 million image pairs, where 131k samples are annotated with editing regions. For those with no editing region annotations, we use them for full-image generation.

We perform training on 4 NVIDIA Tesla A100 GPUs. The model is trained for $3.9M$ steps with a batch size of $100$ and a learning rate of $1e - 4$. Per common practices [58, 60], we evaluate models at a higher image resolution than that used during training (specifically, $512 \times 512$ pixels compared to $256 \times 256$ pixels), and fine-tune them on the target resolution for an additional $2,000$ steps. For the Emu Edit benchmark, we train our model with a learning rate of $1e - 5$ for $60,000$ steps.

### 3.2 Results on Image Editing

**Benchmarks & Evaluation Metrics.** We demonstrate the superiority of our approach on two widely recognized benchmarks: MagicBrush [58] and Emu Edit [36]. The MagicBrush test set provides editing region annotations for each sample, thereby facilitating the evaluation of region-conditioned editing. This benchmark assesses both multi-turn editing, which evaluates the final image after a series of edits, and single-turn editing, which assesses the target image following an individual edit.

The MagicBrush benchmark provides target images and evaluates the similarity between each generated image and the corresponding target image using various metrics, including L1 distance, L2

Table 1: **Results on the MagicBrush test set for region-aware editing.** We compare NEP with existing approaches under single-turn and multi-turn settings with our results labeled in gray.

| Settings | Methods | L1↓ | L2↓ | CLIP-I↑ | DINO↑ |
|---|---|---|---|---|---|
| | *Global Description-guided* | | | | |
| | SD-SDEdit | 0.1014 | 0.0278 | 0.8526 | 0.7726 |
| | Null Text Inversion | 0.0749 | 0.0197 | 0.8827 | 0.8206 |
| | GLIDE | 3.4973 | 115.8347 | 0.9487 | 0.9206 |
| | Blended Diffusion | 3.5631 | 119.2813 | 0.9291 | 0.8644 |
| **Single-turn** | *Instruction-guided* | | | | |
| | HIVE | 0.1092 | 0.0380 | 0.8519 | 0.7500 |
| | InstructPix2Pix (IP2P) | 0.1141 | 0.0371 | 0.8512 | 0.7437 |
| | IP2P w/ MagicBrush | 0.0625 | 0.0203 | 0.9332 | 0.8987 |
| | UltraEdit | 0.0575 | 0.0172 | 0.9307 | 0.8982 |
| | FireEdit | 0.0701 | 0.0238 | 0.9131 | 0.8619 |
| | AnySD | 0.1114 | 0.0439 | 0.8676 | 0.7680 |
| | EditAR | 0.1028 | 0.0285 | 0.8679 | 0.8042 |
| | Ours | **0.0547** | **0.0163** | **0.9350** | **0.9044** |
| | *Global Description-guided* | | | | |
| | SD-SDEdit | 0.1616 | 0.0602 | 0.7933 | 0.6212 |
| | Null Text Inversion | 0.1057 | 0.0335 | 0.8468 | 0.7529 |
| | GLIDE | 11.7487 | 1079.5997 | 0.9094 | 0.8494 |
| | Blended Diffusion | 14.5439 | 1510.2271 | 0.8782 | 0.7690 |
| **Multi-turn** | *Instruction-guided* | | | | |
| | HIVE | 0.1521 | 0.0557 | 0.8004 | 0.6463 |
| | InstructPix2Pix (IP2P) | 0.1345 | 0.0460 | 0.8304 | 0.7018 |
| | IP2P w/ MagicBrush | 0.0964 | 0.0353 | 0.8924 | 0.8273 |
| | UltraEdit | 0.0745 | **0.0236** | 0.9045 | 0.8505 |
| | FireEdit | 0.0911 | 0.0326 | 0.8819 | 0.8010 |
| | AnySD | 0.0748 | 0.0273 | **0.9152** | **0.8623** |
| | EditAR | 0.1341 | 0.0433 | 0.8256 | 0.7200 |
| | Ours | **0.0707** | 0.0269 | 0.9107 | 0.8493 |

distance, CLIP feature similarity (CLIP-I), and DINO feature similarity. Additionally, it measures text-image consistency by comparing the CLIP feature similarity (CLIP-T) between the generated image and the caption of the target image.

The Emu Edit test set does not provide target images; therefore, the evaluation of editing region regeneration is conducted separately from the reconstruction of unedited regions. The regeneration process is assessed using two metrics: CLIP text-image similarity (CLIPout) and CLIP text-image direction similarity (CLIPdir) measure the consistency between the change in images and the change in captions. The reconstruction quality is measured by comparing the edited image to the original source image in terms of L1 distance, CLIP image similarity (CLIPimg), and DINO similarity.

### 3.2.1 Quantitative Results

We demonstrate the superiority of NEP in terms of region-aware editing on the MagicBrush test set. The compared prior arts broadly fall into two categories: (1) global description-based, such as SD-SDEdit [20], Null Text Inversion [21], GLIDE [23], as well as Blended Diffusion [1], and (2) instruction-guided, including HIVE [59], InstructPix2Pix [2], MagicBrush [58], UltraEdit [60], FireEdit [61], AnySD [54] and EditAR [22]. Table 1 demonstrates that our approach achieves the highest score for single-turn editing and better or comparable performance under the multi-turn setting. For the first time, autoregressive models can achieve top performance on well-recognized editing benchmarks.

We demonstrate the effectiveness of free-form editing on the Emu Edit test set [36]. We compare NEP with state-of-the-art approaches including InstructPix2Pix [2], MagicBrush [58], Emu Edit [36] UltraEdit [60], MIGE [44], and AnySD [54]. In Table 2, we can observe that, without resorting to editing masks, our approach still achieves comparable or better editing performance.

Table 2: **Results on Emu Edit Test for free-form editing.** Our approach is highlighted in gray .

| Method | CLIPdir↑ | CLIPout↑ | L1↓ | CLIPimg↑ | DINO↑ |
|---|---|---|---|---|---|
| InstructPix2Pix | 0.0784 | 0.2742 | 0.1213 | 0.8518 | 0.7656 |
| MagicBrush | 0.0658 | 0.2763 | 0.0652 | 0.9179 | **0.8924** |
| Emu Edit | 0.1066 | 0.2843 | 0.0895 | 0.8622 | 0.8358 |
| UltraEdit | **0.1076** | 0.2832 | 0.0713 | 0.8446 | 0.7937 |
| MIGE | 0.1070 | 0.3067 | 0.0865 | 0.8714 | 0.8432 |
| AnyEdit | 0.0626 | 0.2943 | **0.0673** | **0.9202** | **0.8919** |
| Ours | 0.1064 | **0.3078** | 0.0781 | 0.8710 | 0.8440 |

Table 3: **Ablation studies on the MagicBrush test set under the multi-turn setting**. We validate the contribution of each design choice by removing them and observing the performance drop. We ablate two aspects: 1) **ERC** by removing the editing & unediting tokens inferred from editing region masks, and 2) **NEP vs. NTP** by generating full image tokens. The default setting is highlighted in gray .

| Methods | #Output Tokens | L1↓ | L2↓ | CLIP-I↑ | DINO↑ |
|---|---|---|---|---|---|
| NEP | $L_E$ | **0.0712** | **0.0272** | **0.9097** | **0.8459** |
| w/o ERC | $L_E$ | 0.0741 | 0.0281 | 0.9040 | 0.8372 |
| NTP | $L$ | 0.0968 | 0.0309 | 0.8854 | 0.8235 |

### 3.2.2 Ablation Study

We perform ablation studies on the Magicbrush multi-turn test set. For each configuration, we report the results of the models trained for $30,000$ steps. We assess two critical design choices. First, we exclude mask embeddings, relying solely on text and source images as inputs, which degrades performance as shown in Table 3. Qualitatively, we observe that removing ERC increases the likelihood of the model making no changes to the source model, as demonstrated in Figure 3. Second, we remove the next editing token positions by generating all tokens in a raster scan order, following an NTP framework. Without any priors on editing regions, this leads to a significant performance drop, highlighting the need for targeted token generation.

### 3.2.3 Computational efficiency

Table 4 demonstrates comparative results on computational cost. NEP requires higher GPU resources due to the concatenation of mask embeddings along the sequential dimension (Section 2.2), which increases sequence length and attention computational cost. Despite this, our approach achieves the fastest editing speed as we only need to predict editing region tokens rather than the whole image as diffusion models or AR-based models do.

Table 4: Computational cost averaged across MagicBrush test samples.

| Methods | Memory (GB) | Inference time (s) |
|---|---|---|
| UltraEdit | **4.04** | 2.94 |
| EditAR | 6.59 | 10.70 |
| NEP | 13.25 | **2.88** |

### 3.2.4 Qualitative Results

Figure 4 presents qualitative comparisons with state-of-the-art methods. Apart from avoiding unintended modifications to the input image, as shown in Figure 1, our approach excels in following the provided instructions to perform faithful and accurate modifications. Additionally, it is capable of making fine-grained modifications (e.g., changing the outfit), showcasing its high versatility and precision in handling complex editing instructions.

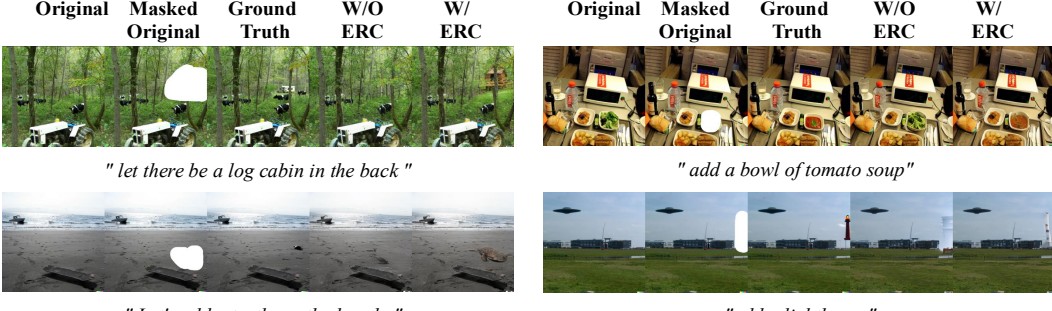

| Original Masked Original | Ground Truth | W/O ERC | W/ ERC | | Original Masked Original | Ground Truth | W/O ERC | W/ ERC |

*" let there be a log cabin in the back "*       *" add a bowl of tomato soup"*

*" Let's add a turtle on the beach. "*       *"add a lighthouse"*

Figure 3: **Visualized ablation on ERC.** This demonstrates that removing Editing Region Conditioning increases the editing model's change to refuse to modify the source image. Best viewed zoomed in and in color.

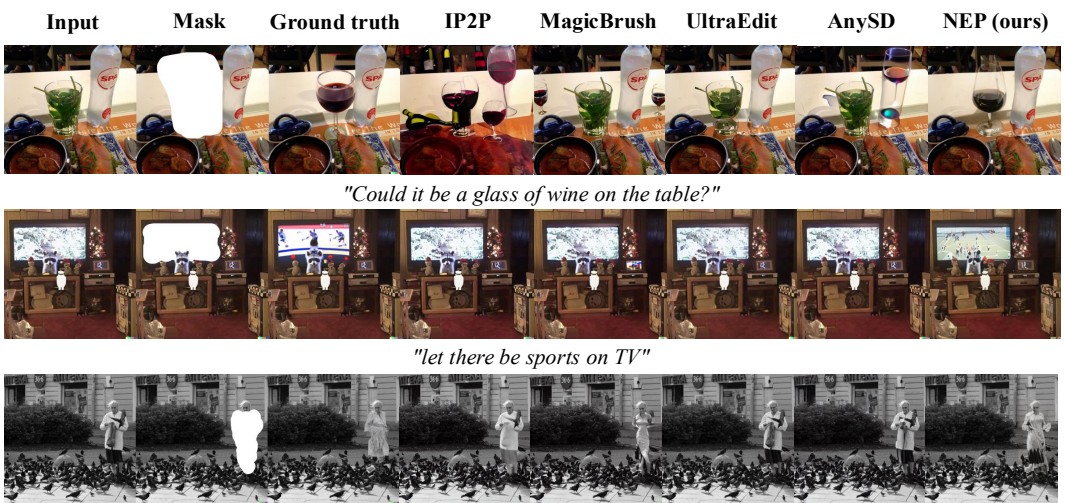

| Input | Mask | Ground truth | IP2P | MagicBrush | UltraEdit | AnySD | NEP (ours) |

*"Could it be a glass of wine on the table?"*

*"let there be sports on TV"*

*"let the woman wear a designer gown"*

Figure 4: **Comparative editing results.** This demonstrates that our approach can make more faithful edits to source images, either by updating objects (case #1, #2), or making fine-grained edits (case #3). Best viewed zoomed in and in color.

## 3.3 Results on NEP Pre-training

To better understand how NEP works, we also evaluate the intermediate text-to-image model RLlamaGen obtained during NEP pretraining. RLlamaGen acquires the zero-shot editing ability without sacrificing text-to-image generation performance.

### 3.3.1 Zero-shot Image Editing

We demonstrate that RLlmaGen is readily capable of image editing. This is achieved by regenerating tokens in the editing regions. Figure 5 demonstrates that RLlmaGen can make fine-grained and coherent edits.

**Comparison with Localized Editing Approaches.** We compare our zero-shot editing performance against aMUSEd [27], which is also capable of localized zero-shot editing. We use its publicly available checkpoint for comparison, adhering to its default configu-

Table 5: **Comparative Results on Zero-shot Editing on MagicBrush test set.**

| Settings | Methods | L1↓ | L2↓ | CLIP-I↑ | DINO↑ |
|---|---|---|---|---|---|
| Single-turn | aMUSEd | 0.0913 | 0.0300 | 0.8802 | 0.8131 |
| | Ours (zero-shot) | **0.0743** | **0.0211** | **0.9032** | **0.8509** |
| Multi-turn | aMUSEd | 0.1034 | 0.0361 | 0.8689 | **0.8092** |
| | Ours (zero-shot) | **0.0916** | **0.0319** | **0.8798** | 0.7859 |

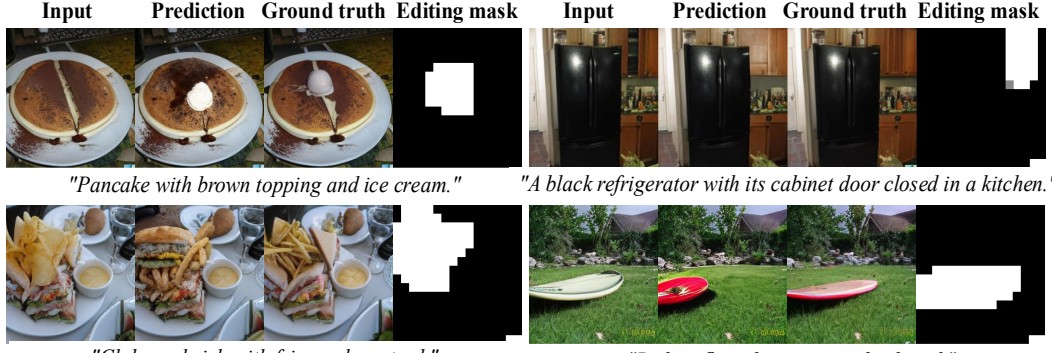

| Input | Prediction | Ground truth | Editing mask | Input | Prediction | Ground truth | Editing mask |

*"Pancake with brown topping and ice cream."*      *"A black refrigerator with its cabinet door closed in a kitchen."*

*"Club sandwich with fries and mustard."*      *"Red surfboard on grass in backyard."*

Figure 5: **Examples of RLlamaGen's zero-shot editing capability.** It can make fine-grained edits such as adding external objects (ice cream in example #1), changing the state of input objects (cabinet door open → closed in example #2), changing the semantics (chips → fries in example #3), and changing the color (white → red in example #4). Best viewed zoomed in and in color.

rations[1]. Results on the MagicBrush dataset show that our approach outperforms aMUSEd. This is attributed to our method's ability to enable fine-grained editing by keeping all source image tokens visible to the generation model, whereas aMUSEd replaces edited regions with mask tokens, limiting its precision.

**Ablations on Generation Order.** Alternative to the default generation order, i.e., an in-mask raster scan order, as we introduced in Section 3.3, we employ random generation order for zero-shot image editing. The results in Table 6 demonstrate that altering the generation order has negligible impact on the effectiveness of our approach, confirming its robustness.

Table 6: **Ablations on Generation Order for Zero-shot Editing on MagicBrush test set.**

| Settings | Methods | L1↓ | L2↓ | CLIP-I↑ | DINO↑ |
|---|---|---|---|---|---|
| **Single-turn** | In-mask random order | **0.0741** | **0.0211** | 0.9027 | 0.8482 |
| | In-mask raster scan order | 0.0743 | **0.0211** | **0.9032** | **0.8509** |
| **Multi-turn** | In-mask random order | **0.0911** | **0.0316** | 0.8782 | 0.7833 |
| | In-mask raster scan order | 0.0916 | 0.0319 | **0.8798** | **0.7859** |

### 3.3.2 Text-to-Image Generation Results

**Benchmarks & Evaluation Metrics.** We evaluate the image generation quality on MS-COCO 30K in terms of Fréchet Inception Distance (FID) and CLIP similarity. FID reflects the fidelity and diversity of generated images. It measures the distance between the ground truth image distribution and the generated image distribution, where the distributions are constituted of Inception V3 [43] embeddings extracted from corresponding images. The CLIP score is used to evaluate the instruction-following ability of T2I models. It measures the similarity between the vision embeddings extracted from the generated image and text encoder embeddings extracted from corresponding captions.

We demonstrate that randomized pre-training preserves raster scan generation capability. Moreover, employing NEP test-time scaling further improves generation performance. Table 7a shows that RLlamaGen outperforms its baseline (line 2 vs. line 1), and performs similarly with LlamaGen tuned for the same number of steps (line 2 vs. line 3). Scaling NEP for self-refinement can obtain 1.5% improvement in terms of CLIP and 11.4% reduction in FID (line 4 vs. line 2).

### 3.4 Results on Test-time Scaling of NEP

We evaluate our self-improvement strategy on top of NEP, which iteratively revises the model's previous generation. This self-improvement can be effectively scaled through multi-round iterative refinement. Empirical evidence suggests that masking out previously generated tokens during the revision process yields superior results; thus, we adopt this approach as our default method.

We demonstrate the scaling effects of NEP in Table 7b, where we observe consistent improvements as the number of revision rounds increases. This strategy can be further enhanced by utilizing stronger

---

[1]https://huggingface.co/blog/amused

| Original | Round 1 | Round 2 | Round 3 | Original | Round 1 | Round 2 | Round 3 |

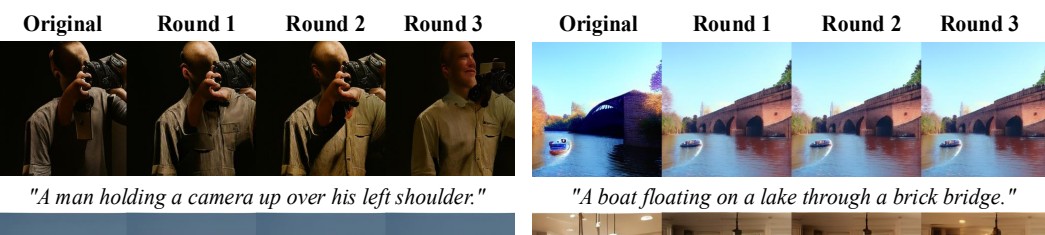

*"A man holding a camera up over his left shoulder."*  |  *"A boat floating on a lake through a brick bridge."*

*"A passenger plane that is parked on the runway."*  |  *"A kitchen filled with lots of counter top space."*

Figure 6: **Self-improving RLlamaGen.** By gradually revising the original output, we can obtain images better aligned with instructions and with higher fidelity. Best viewed zoomed in and in color.

Table 7: **Results on NEP pre-training and TTC.** The pre-trained RLlamaGen enables arbitrary order generation without sacrificing generation quality. NEP can be employed for test-time scaling which enhances the generation further.

(a) **Pretraining schemes**. Comparative results between LlamaGen baseline, RLlamaGen fine-tuned for a pre-defined number of steps, and LlamaGen fine-tuned for the same number of steps.

| Methods | CLIP↑ | FID↓ |
|---|---|---|
| LlamaGen | 0.320 | 15.07 |
| LlamaGen ft. | 0.326 | 12.00 |
| RLlamaGen | 0.325 | 11.49 |
| TTS w/ NEP | **0.330** | **10.18** |

(b) **Test-time scaling w/o post-training.** NEP can be used to iteratively revise generated images. The generation quality gradually improves and saturates after 2 iterations.

| # Revision rounds | CLIP↑ | FID↓ |
|---|---|---|
| 0 | 0.325 | 11.49 |
| 1 | **0.332** | 9.94 |
| 2 | **0.332** | 9.93 |
| 3 | **0.332** | 9.85 |
| 4 | **0.332** | **9.82** |

verifier models and training the model for self-improvement. The revision process is visualized in Figure 6, showcasing better alignment with the conditioning text prompts and higher fidelity.

# 4 Related Works

## 4.1 Text-to-Image Generation

Text-to-image generation has become a cornerstone of modern artificial intelligence, enabling to create visual content based on textual descriptions. Pioneering models such as Generative Adversarial Networks (GANs) [10] make groundbreaking breakthroughs by generating high-fidelity images. AttnGAN [52] built on StackGAN [57] achieves better alignment with text instructions. However, GANs still faced challenges like training instability (e.g., mode collapse, where the model generates limited varieties of images) and difficulty with highly detailed or multi-object scenes, setting the stage for the next evolutionary step.

More recently, diffusion models [37, 13, 38] like Stable Diffusion [32] have emerged, creating realistic images by iteratively denoising random noise guided by text descriptions, setting a new standard for quality and versatility. However, the learning paradim and architectures diverge from well-established large language models (LLMs) [3],, making it difficult for artificial general intelligence featuring a shared framework for various modalities.

In this regard, a line of works [28, 29, 53] resort to autoregressive models for visual generation. Images are tokenized into a sequence of tokens and generated sequentially based on prefilled text tokens. Benefiting from large-scale models and datasets, they can create photorealistic images with a remarkable text-following capability. This field is further advanced by several open-source works, such as LlamaGen [42], Emu3 [48], and Janus [49].

## 4.2 Image Editing

Image editing builds on text-to-image generative models by conditioning outputs on source images, but preserving unedited regions poses a challenge for diffusion models. These models require looking for mapping latent representations for the original RGB values, often using inversion techniques [39, 21]. However, such methods typically demand inference-time tuning, such as tuning textual embeddings [9], model weights [33, 47], or null-text embeddings [21] to enable classifier-free guidance [12]. Even when noise trajectories across varying levels are available, maintaining unedited regions is not assured. For instance, Prompt-to-Prompt [11] introduces a time threshold to prioritize generating target object geometry through text-to-image steps without source image conditioning, trading off reconstruction accuracy for generative flexibility.

Efforts to guide edits using user-specified masks have been explored in both training-free [1] and training-based approaches [23, 58, 60]. Training-free methods apply masks across all diffusion steps to blend source image latents with text-conditioned outputs, while training-based methods append an extra channel to the source image for guidance. Despite these advancements, both approaches require full image regeneration, which hampers efficiency during training and inference.

In contrast, our work enables localized editing by regenerating tokens solely within user-defined regions, preserving pixels outside these areas without modification. Leveraging user-provided masks introduces minimal limitations, thanks to recent advances in segmentation techniques [15, 30, 16].

## 4.3 Test-time Scaling for Text-to-Image Generation

The success of LLMs' inference-time scaling motivates the exploration of similar behavior for text-to-image generation. Existing approaches mainly investigate diffusion model scaling, either by increasing the denoising step [14, 41] or employing best-of-N sampling [19]. More recently, new test-time scaling approaches have emerged that enable revising prior generations by incorporating corrections and feedback into the context [17]. However, an additional post-training stage is required to support their iterative refinement, limiting their flexibility and increasing computational demands. In this work, we investigate inference-time scaling in autoregressive image generation models that can conduct self-improvement utilizing NEP, offering a new perspective on enhancing model performance during testing without dedicated post-training.

# 5 Conclusion

In this work, we propose a next-editing token-prediction pipeline for text-driven image editing. It allows for easy localized editing without making unintended modifications to the non-editing region. To support regeneration at any user-specified position, we pre-train an any-order autoregressive T2I model that can generate tokens in arbitrary orders. Furthermore, we demonstrate NEP can be integrated into an iterative refinement loop for test-time scaling.

# 6 Limitations and Broader Impacts

**Limitations.** While the proposed approach demonstrates promising results, it relies on user-provided masks for guidance to prevent unintended modifications to the source image. This requirement adds extra computation or annotation, making the process less efficient. We plan to address automated and unified masking region localization in future work. Additionally, the robustness of Neural Editing Propagation (NEP) to noise in editing region masks remains uncertain. Imperfect user-specified masks lead to two primary scenarios: 1) the segmentation mask is larger than the ground truth editing region, and 2) the segmentation mask is smaller. In the first scenario, NEP exhibits robustness, achieving comparable results, as shown in Table 2 for free-form image editing without a mask. In the second scenario, our approach lacks specific optimization. We plan to develop a pipeline for automatically refining user-specified masks in future work.

**Social impacts.** Our primary motivation for developing image editing algorithms is to foster innovation and creativity; however, we recognize that they also present significant ethical and societal challenges. We are committed to minimizing these risks by filtering training images for unsafe content and restricting the model's use to research purposes only upon release. In the future, we will actively engage in discussions and initiatives aimed at mitigating these risks.

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
