# OpenReview forum: "NEP: Autoregressive Image Editing via Next Editing Token Prediction"
_NeurIPS.cc/2025/Conference — NeurIPS 2025 poster_

### Official Review · Reviewer_KK48 · 2025-07-01

**Clarity:** 2
**Significance:** 2
**Originality:** 2
**Rating:** 5
**Confidence:** 4

**Summary:**

This paper introduces NEP (Next Editing-token Prediction) for text-guided image editing that uses an autoregressive (AR) model. The main goal for introducing this framework is to reduce unnecessary computational costs and unintended changes in non-edited regions. To enable editing of arbitrary regions, the method first involves pre-training RLlamaGen to generate image tokens in any user-specified order. This is achieved by incorporating shuffled positional embeddings during training. The model is then fine-tuned for the editing task, where it learns to use text instructions, a source image, and a mask to predict the new tokens for the editing region.

**Questions:**

1. Clarification on test-time scaling models: Could you provide the implementation details for the critic and reward models used in Section 2.3? What models were used, and how were they trained?
2. Questionable baseline results: In Table 1, the L1 and L2 distance metrics for GLIDE and Blended Diffusion are several orders of magnitude worse than all other methods. Could you clarify if these models were implemented and evaluated correctly, as the results appear to be outliers?
3. Missing comparisons: Why were other relevant token-based editing methods like MaskGIT or Painter, and other randomized autoregressive models like RAVG, not included in the quantitative comparisons in your tables?
4. Inference latency: Could you provide a quantitative comparison of the inference time for a typical edit using NEP versus a leading diffusion model like InstructPix2Pix? This would provide concrete evidence for the claimed efficiency benefits.

**Ethical Concerns:**

["NO or VERY MINOR ethics concerns only"]

**Final Justification:**

I have read the authors' responses to my questions and concerns. The response has nicely addressed my existing concerns and therefore I increase my score from 4 to 5.

**Limitations:**

The authors are up-front about the method's heavy reliance on user-provided masks, correctly identifying that this adds an extra layer of computation or annotation that impacts the overall efficiency of the pipeline.

**Paper Formatting Concerns:**

Not of a concern I am aware of.

**Quality:**

2

**Strengths And Weaknesses:**

### Strengths

* The paper demonstrates that NEP can be used for test-time scaling, where the model iteratively refines its own output by identifying and regenerating low-quality regions.

* This approach achieves state-of-the-art results on standard image editing benchmarks like MagicBrush.

* The idea of NEP is technically sound and the paper has successfully demonstrated its capabilities.

***

### Weaknesses

* Incomplete experimental comparisons: The paper fails to compare NEP against other highly relevant models. The comparisons would be stronger if they included other token-based editing methods like MaskGIT [1] or Painter [2], and other randomized autoregressive models like RAVG [3].
* No inference time analysis: A core claim of the paper is improved efficiency by avoiding full-image generation. However, the paper provides no quantitative data, such as an inference latency comparison against diffusion-based methods, to support this claim.
* Underwhelming performance on free-form editing: On the Emu Edit benchmark, NEP's performance is not state-of-the-art. It is outperformed by other methods like AnyEdit and MagicBrush on most metrics, including L1 distance, CLIPimg similarity, and DINO similarity. This suggests the method is less effective without the strong guidance of a mask.
* Marginal improvement from test-time scaling: The results in Table 4b show that while one round of revision offers a noticeable improvement, the gains from further iterations are minimal and can even degrade performance, as round 3 is worse than round 2. This raises questions about the robustness and practical value of multi-round iterative refinement.
* Heavy reliance on user-provided masks: The method's precision is contingent on having an accurate editing mask. The authors acknowledge this as a limitation, as it requires either manual annotation or an external segmentation model, adding an extra step and computational overhead to the pipeline.
* Lack of implementation details:
    * The paper does not specify the implementation of the "critic model" and "reward model" used for test-time scaling in Section 2.3. This omission makes the test-time scaling component difficult to reproduce.
    * Figure 2 does not clearly illustrate how the text, source image, and mask embeddings are combined and processed by the AR model to generate the final output.

### References

[1] Chang, H., Zhang, H., Lu, J., Koss, M., Wang, X., & Zeng, A. (2022). MaskGIT: Masked Generative Image Transformer. In *Proceedings of the IEEE/CVF Conference on Computer Vision and Pattern Recognition*.

[2] Wang, X., Zhang, H., He, T., Chen, X., & Liu, S. (2023). Painter: A generalist model for in-context understanding of images. *arXiv preprint arXiv:2305.10657*.

[3] Yu, Q., He, J., Deng, X., Shen, X., & Chen, L. C. (2024). Randomized autoregressive visual generation. *arXiv preprint arXiv:2411.00776*.

---

> ### Author Rebuttal · Authors · 2025-07-30
>
> We thank the reviewer for recognizing the strengths of our approach, including its solidness and capabilities, and for providing valuable suggestions. We revised the manuscript as follows:
> - Added comparisons with related works (W1, Q3), expanded free-form editing results (W3), and updated test-time scaling results (W4).
> - Compared inference speed with state-of-the-art editing methods (W2, Q4).
> - Discussed limitations further (W5).
> - Included more implementation details (W6, Q1) and expanded MagicBrush results explanation (Q2).
>
> In the following, we respond to the reviews point-by-point.
>
> > W1. Incomplete experimental comparisons: ...MaskGIT [1] or Painter [2], and RAVG [3].
>
> We sincerely appreciate the reviewer's suggestion. We agree that including more comparative results is beneficial to the solidness of our work. We discuss the comparison with the mentioned approaches individually.
> 1. Comparison with MaskGIT
>
> We would like to respectfully clarify that a direct comparison with MaskGIT is not feasible, as it only supports class-conditional editing. Instead, we use its extension, MUSE [2], which supports text-prompted generation. However, MUSE's official code is not open-sourced, and reproduced implementations, such as “lucidrains/muse-maskgit-pytorch” and “huggingface/open-muse,” lack pre-trained checkpoints. To address this, we adopt aMUSEd [3], an open-source reproduction of MUSE. This approach has been integrated into the Diffusers library and we use its default configurations as outlined in its official Hugging Face blog. For a fair comparison, we compare with our zero-shot model (Section 2.1). Our results on the MagicBrush dataset show that our zero-shot approach outperforms aMUSEd. This is attributed to our method’s ability to enable fine-grained editing by keeping all source image tokens visible to the generation model, whereas aMUSEd replaces edited regions with mask tokens, limiting its precision. Please see the following for a detailed comparison.
>
> | Settings | Methods | L1↓ | L2↓ | CLIP-I↑ | DINO↑ |
> | --- | --- | --- | --- | --- | --- |
> | Single-turn | Amused         | 0.0913 | 0.0300 | 0.8802 | 0.8131 |
> | Single-turn | Ours (zero-shot) | 0.0743 | 0.0211 | 0.9032 | 0.8509 |
> | Multi-turn | Amused         | 0.1034 | 0.0361 | 0.8689 | 0.8092 |
> | Multi-turn | Ours (zero-shot)| 0.0916 | 0.0319 | 0.8798 | 0.7859 |
>
> 2. Comparison with Painter
>
> We wish to respectfully clarify that using Painter for image editing is challenging and costly because it does not accept explicit task prompts (e.g., “add a mirror to the wall”) for inference. Instead, it requires task prompts to be defined through input-output image pairs. This process is both complex and limiting, as suitable image pairs are often unavailable. In our analysis, out of 1,053 instructions in the MagicBrush test set, only 36 have corresponding input-output pairs in our UltraEdit training set (comprising 4 million pairs). To enable a fair comparison, we evaluate Painter on this subset and compare against NEP’s zero-shot inference results on the same subset. Due to the absence of complete multi-turn instruction sequences in the training data, we conducted evaluations solely under a single-turn setting. The results in the table below demonstrate that Painter significantly lags behind NEP.
>
> | Settings | Methods | L1↓ | L2↓ | CLIP-I↑ | DINO↑ |
> | --- | --- | --- | --- | --- | --- |
> | Single-turn | Painter         | 0.1443 | 0.0602 | 0.7999 | 0.6383 |
> | Single-turn | Ours (zero-shot) | 0.0797 | 0.0198 | 0.8942 | 0.8573 |
>
> This performance gap is primarily due to the mismatch between Painter’s training tasks and image editing. It is trained on visual understanding (e.g., depth estimation, semantic segmentation, human keypoint detection, panoptic segmentation) and low-level image processing tasks (e.g., image denoising, deraining, enhancement), which differ substantially from image editing. Although it claims generalization to out-of-domain tasks (e.g., open-category object segmentation, keypoint detection, instance segmentation), these tasks, however, are limited to vision understanding. When transferred to a high-level generative task, i.e., image editing, it struggles to generate effectively. The edited images produced by Painter reveal that it often disregards the task prompt specified by the input-output image pair and applies one of its learned tasks, such as image segmentation or denoising, to process the input image. Thus, we can conclude that NEP offers superior flexibility and performance.
>
> 3. Comparison with RAVG
>
> We compare our approach with RAVG, adapted for text-to-image generation as it was originally developed for class-to-image generation. The results are presented below.
>
> | Settings | Methods | L1↓ | L2↓ | CLIP-I↑ | DINO↑ |
> | --- | --- | --- | --- | --- | --- |
> | Single-turn | RAVG adapted for T2I | 0.0711 | 0.0188 | 0.9203 | 0.8888 |
> | Single-turn | Ours                 | 0.0547 | 0.0163 | 0.9350 | 0.9044 |
> | Multi-turn | RAVG adapted for T2I  | 0.0968 | 0.0309 | 0.8854 | 0.8235 |
> | Multi-turn | Ours                  | 0.0707 | 0.0269 | 0.9107 | 0.8493 |
>
> NEP demonstrates superior performance over the adapted RAVG for image editing. This advantage stems from NEP’s optimization for editing and regeneration tasks exclusively, whereas RAVG must balance regeneration and reconstruction, resulting in compromised editing performance.
>
> > W2. No inference time analysis...
>
> We thank the reviewer for raising this insightful question, as validating the inference efficiency is crucial to our contribution. We have listed these numbers below:
>
> |Methods| Memory (GB) | Inference time (s) |
> |-|-|-|
> |UltraEdit|4.04|2.94|
> |EditAR|6.59|10.70|
> |NEP|13.25|2.88|
>
> NEP requires higher GPU resources due to the concatenation of mask embeddings along the sequential dimension (Section 2.2), which increases sequence length and attention computational cost. Despite this, our approach achieves the fastest editing speed as we only need to predict editing region tokens rather than the whole image as AR-based models do. We have included this analysis in the revised manuscript.
>
> > W3. Underwhelming performance on free-form editing...
>
> We sincerely appreciate the reviewer’s thoughtful feedback and the opportunity to clarify our results on the Emu Edit benchmark. We acknowledge the reviewer’s observation regarding NEP’s performance on metrics such as L1 distance, CLIPimg similarity, and DINO similarity. To provide further context, we note that these metrics measure alignment between the generated and source images, which can be less indicative of editing quality where ground-truth post-editing images are unavailable. Lower scores on these metrics may reflect a model’s tendency to copy the source image rather than editing effectiveness. In our revised manuscript, we’ve emphasized metrics like CLIPdir (consistency between caption and image changes) and CLIPout (alignment with the target caption), where NEP achieves comparable scores and, in the case of CLIPout, the highest performance. To address the reviewer’s valuable feedback, we’ve expanded our analysis in the manuscript to better explain the nuances of these metrics and their implications for free-form editing. We are deeply grateful for your insights, which have helped us strengthen our evaluation, and we warmly welcome any further suggestions to refine our work.
>
>
>
> > W4. Marginal improvement from test-time scaling...
>
> We thank the reviewer for pointing this out. During the period between submitting our original manuscript and the rebuttal phase, we identified and corrected a bug in our test-time scaling experiment by refining predictions based on the previous round rather than the initial round. Accordingly, we have updated the results in Table 4b of the revised manuscript to reflect these corrections, as detailed below. We can observe that consistent improvement with increasing revision rounds. We appreciate the reviewer’s input and believe this revision strengthens the reliability of our findings.
>
>
> |# Revision rounds | CLIP ↑ | FID ↓ |
> | - | - | - |
> | 0 | 0.325 | 11.49 |
> | 1 | 0.330 | 10.31 |
> | 2 | 0.332 | 9.93 |
> | 3 | 0.332 | 9.85 |
> | 4 | 0.332 | 9.82 |
>
>
>
> > W5. Heavy reliance on user-provided masks...
>
> Thank you for your thoughtful question. We acknowledge that the current reliance on user-provided masks introduces extra constraints to NEP's applicability, as we have discussed in our limitation section. However, we would like to respectfully point out that it is a common requirement among existing image editing methods, including Blended Diffusion, GLIDE, InstructPix2Pix, UltraEdit, and MIGE. Despite the constraint, we have evaluated NEP in a free-form editing setting (Table 2) and achieved competitive results. To address this concern, we have expanded the discussion in the revised manuscript to include our ongoing efforts to minimize mask dependency and enhance free-form editing capabilities. We greatly appreciate your feedback, as it helps guide our future work toward more seamless and intuitive editing tools.
>
>
> > W6. Lack of implementation details
>
> We thank the reviewer for the valuable feedback and apologize for any confusion. To clarify, we utilize the off-the-shelf CLIP-ViT-B/32 as the critic model and ImageReward as the reward model. These details have been explicitly included in the revised manuscript. Additionally, the text, source image, and mask are tokenized and sequentially concatenated as conditioning for the autoregressive model. We have revised Figure 2 to enhance clarity. We appreciate your input, which has strengthened our presentation.
>
> > Q2. Questionable baseline results: ...GLIDE and Blended Diffusion...
>
> Thank you for your insightful question. These numbers are released along with MagicBrush paper (Table 3).
>
> [1] Chang, Huiwen, et al. Muse: Text-To-Image Generation via Masked Generative Transformers. ICML 2023.
>
> [2] Patil, Suraj, et al. amused: An open muse reproduction. arXiv 2024.

---

### Official Review · Reviewer_9n1p · 2025-07-02

**Clarity:** 3
**Significance:** 3
**Originality:** 4
**Rating:** 5
**Confidence:** 4

**Summary:**

This paper presents a novel efficient way of modeling text-guided image editing task with autoregressive transformers and mask conditioning. By predicting “what really needs to change” rather than “the whole image,” it delivers controllable, fine-grained edits with higher faithfulness and markedly better compute efficiency, and it inherits the natural test-time scalability for further quality gains at inference time.

**Questions:**

1. please address weakness points
2. Are output editing tokens always predicted in raster scan order? Since the positional embedding corresponding to the
next token in the assigned order is added, can we use arbitrary order for output editing tokens?

**Ethical Concerns:**

["NO or VERY MINOR ethics concerns only"]

**Final Justification:**

The rebuttal addresses my concerns: (1) it resolves my major concern about the effectiveness of mask embeddings by detailing the two-stage masking mechanism and its performance impact; and (2) it addresses a minor concern by providing complementary results for alternative output orders, showing that random-order pretraining supports arbitrary-order generation at inference. Accordingly, I maintain my scores.

**Limitations:**

yes

**Quality:**

4

**Strengths And Weaknesses:**

Strengths:
1. Any-order next-token editing. Reframes text-guided image editing as an any-order next-token prediction task, enabling the autoregressive transformer to flexibly generate visual tokens in arbitrary sequences rather than a fixed raster scan.

2. Explicit spatial control via mask tokens. Introduces a pair of lightweight “edit/keep” mask tokens that encode the user-specified region, giving the model precise, learnable awareness of where changes are permitted.

3. Variable-length, region-specific generation. By combining (1) and (2), the model produces only the subset of image tokens that fall inside the mask—of whatever shape or size—eliminating redundant computation on untouched areas and supporting fine-grained, mask-conditioned edits.

Weaknesses:
1. In table 3, the performance drop due to removing mask embeddings is minor. What are the intuitions behind this? Does it mean mask embeddings are redundant?
2. What output orders are used for evaluating zero-shot image editing of RLlmaGen? Since it's trained on random order, it's interesting to see how it performs using different orders.

---

> ### Author Rebuttal · Authors · 2025-07-30
>
> We thank the reviewer for recognizing the novelty, flexibility, and effectiveness of our approach. Your support of our work is sincerely appreciated. Below, we provide detailed replies to your comments and hope we can resolve your major concerns.
>
>
> > W1. In table 3, the performance drop due to removing mask embeddings is minor. What are the intuitions behind this? Does it mean mask embeddings are redundant?
>
> Thank you for your constructive comment. We agree that expanding the discussion on the ablation study strengthens our work. The editing region mask is used by NEP twice. First, it determines which token to be predicted by NEP (we feed the "probes" corresponding to the position within the mask region to NEP). Second, we use them on the input to inform the model of editing regions during computation. In Table 3, we incrementally remove the second use (line 1 to line 2) and the first use (line 2 to line 3), observing a more substantial performance drop after removing the first use, suggesting that the first application is more critical. We have clarified this in the revised manuscript to avoid confusion. However, we do not consider the second approach redundant. Our qualitative ablation study suggests that omitting the input mask embedding condition increases the likelihood of the model making no changes to the source model. Due to this year's rebuttal policy prohibiting the sharing of visualized results, we have included them in the revised manuscript. We greatly appreciate your valuable input and welcome further suggestions to enhance our discussion.
>
>
>
> > W2. What output orders are used for evaluating zero-shot image editing of RLlmaGen? Since it's trained on random order, it's interesting to see how it performs using different orders.
>
> Thank you for your question. The zero-shot image editing experiment, illustrated in Figure 4, employs the default generation order, i.e., an in-mask raster scan order, as we introduced in Section 3.3. Specifically, we filter the raster scan tokens, retaining only those in the editing region, and generate them sequentially. In response to your suggestion, we conduct a comparison with a random generation order for the editing tokens. The results are listed in the following table:
>
>
> | Settings | Methods | L1↓ | L2↓ | CLIP-I↑ | DINO↑ |
> | --- | --- | --- | --- | --- | --- |
> | Single-turn | Ours (In-mask raster scan order) | 0.0743 | 0.0211 | 0.9032 | 0.8509 |
> | Single-turn | Ours (In-mask random order) | 0.0741 | 0.0211 | 0.9027 | 0.8482 |
> | Multi-turn | Ours (In-mask raster scan order)| 0.0916 | 0.0319 | 0.8798 | 0.7859 |
> | Multi-turn | Ours (In-mask random order)| 0.0911 | 0.0316 | 0.8782 | 0.7833 |
>
> The results demonstrate that altering the generation order has negligible impact on the effectiveness of our approach. We thank you for your valuable feedback and have incorporated this analysis into the revised manuscript to further clarify our method’s robustness.
>
> > Q1. Are output editing tokens always predicted in raster scan order? Since the positional embedding corresponding to the next token in the assigned order is added, can we use arbitrary order for output editing tokens?
>
> Thank you for your insightful question. Please refer to our response to W2 for details.

---

> > ### Comment · Reviewer_9n1p · 2025-08-02
> >
> > I appreciate the authors’ efforts in providing comprehensive responses.
> > W1. The detailed clarification of the two-stage application of embedding masks clearly demonstrates the effect of the mask embeddings.
> > W2. The additional results using alternative output orders are a valuable complement.
> > Thank you—the responses address my questions.

---

> > > ### Author Response · Authors · 2025-08-05
> > >
> > > Dear Reviewer,
> > >
> > >
> > > Thank you for your thorough and insightful review, which has been crucial to improving our work.
> > >
> > > Best regards,
> > >
> > > The Authors

---

### Official Review · Reviewer_Zgfx · 2025-07-03

**Clarity:** 1
**Significance:** 2
**Originality:** 2
**Rating:** 4
**Confidence:** 4

**Summary:**

This paper introduces NEP (Next Editing-token Prediction), a novel framework for text-guided image editing that avoids regenerating the entire image and instead focuses only on the user-specified editing regions. Built on a pre-trained, arbitrary-order autoregressive model called RLlamaGen, NEP enables efficient, high-fidelity localized edits while preserving non-edited areas. It uses editing masks and random-order token generation to target only relevant regions, reducing computational cost and unintended changes. The proposed method achieves state-of-the-art performance on major benchmarks like MagicBrush and Emu Edit.

**Questions:**

Q1. Could the authors explore alternative strategies for NEP beyond the one described in Eq. (3)? For instance, are there different methods to represent or encode the editing regions that might yield better performance or flexibility?

Q2. For further questions and concerns, please refer to the Major and Minor Weaknesses sections above.

**Ethical Concerns:**

["NO or VERY MINOR ethics concerns only"]

**Final Justification:**

During the rebuttal, my major concerns regarding (1) the novelty of the work and (2) the comparison with AR-based image editing baselines were resolved. In addition, the authors provided promising results on computational cost and discussed improvements in paper writing and presentation, which further alleviated my concerns. Therefore, I increase my score to 4.

**Limitations:**

Yes

**Quality:**

2

**Strengths And Weaknesses:**

**Strengths**

S1. The paper demonstrates strong experimental results; NEP consistency outperforms a variety of baseline methods across several quantitative metrics, as shown in Table 1. This indicates the effectiveness of the proposed method.

S2. The introduction of a test-time scaling strategy is another strength of a paper, as it provides a practical mechanism for iterative self-improvement in image quality without additional training.

**Major Weakness**

W1. Absence of a direct comparison with other autoregressive image editing methods: Given the autoregressive nature of NEP, such comparisons are essential and would greatly strengthen the empirical validation.

W2. The novelty of the proposed method is somewhat limited. It remains unclear how NEP fundamentally differs from prior works that leverage autoregressive methods for image editing, and this lack of distinction weakens the contribution.

W3. The paper lacks an analysis of the computational overhead associated with the proposed method. Important details, such as memory usage (in GB), training and inference runtimes (in seconds), should be explicitly mentioned to quantitatively assess efficiency of NEP.


**Minor Weakness**

W4. The overall writing and presentation are unclear in several parts, making it hard to understand the main strategy and contributions of the paper. A more detailed and intuitive explanation of the NEP framework, supported by a well-structured figure (i.e. improving Figure 2 for intuitively explaining Eq. (4)), would significantly enhance clarity.

W5. The organization of the paper could be improved. For example, moving the related work section into Section 2 would provide a more logical flow, since the related work section is essential to fully understand the method. Also, distincting the preliminaries and the proposed method section would help readers to better understand the paper.

---

> ### Author Rebuttal · Authors · 2025-07-30
>
> We greatly appreciate the reviewer’s insightful and valuable comments. Thank you for your positive feedback on the novelty and empirical effectiveness of NEP. To address your concerns, we have added comparative experiments to evaluate effectiveness (W1) and efficiency (W3). We have also expanded discussions on the distinctions of this work (W2) and our design choices (Q1). Additionally, we have revised the manuscript to improve clarity and writing quality (W4) and enhance overall organization (W5). Below, we address your comments individually.
>
>
> > W1. Absence of a direct comparison with other autoregressive image editing methods: Given the autoregressive nature of NEP, such comparisons are essential and would greatly strengthen the empirical validation.
>
> Thank you for your valuable suggestion. We agree that including more comparative results with other autoregressive image editing methods can make our work stronger. In response to your feedback, we have conducted a comparative experiment with EditAR[1], an autoregressive image editing approach accepted by CVPR 2025. EditAR was not included in the original manuscript as it was not accepted at the time of our initial submission. In response to your feedback, we have conducted a comparative experiment on the Magicbrush dataset for the revised version. We use EditAR’s publicly available checkpoint, inference code, and default hyperparameters for the editing task. The results are presented below:
>
> | Settings | Methods | L1↓ | L2↓ | CLIP-I↑ | DINO↑ |
> | --- | --- | --- | --- | --- | --- |
> | Single-turn | EditAR | 0.1028 | 0.0285 | 0.8679 | 0.8042 |
> | Single-turn | Ours | 0.0547 | 0.0163 | 0.9350 | 0.9044 |
> | Multi-turn | EditAR | 0.1341 | 0.0433 | 0.8256 | 0.7200 |
> | Multi-turn | Ours | 0.0707 | 0.0269 | 0.9107 | 0.8493 |
>
> Our approach demonstrates significantly improved performance. This advantage stems from NEP’s optimization, which focuses exclusively on editing and regeneration tasks. In contrast, EditAR must balance both regeneration and reconstruction, which can compromise its editing performance. We have included these comparative results and clarified this distinction in the revised manuscript, to better highlight NEP’s strengths. We appreciate the reviewer’s input and welcome further suggestions to refine our analysis.
>
>
> Please kindly refer to our response to Reviewer KK48 W1 for more comparative experiments.
>
>
> > W2. The novelty of the proposed method is somewhat limited. It remains unclear how NEP fundamentally differs from prior works that leverage autoregressive methods for image editing, and this lack of distinction weakens the contribution.
>
> We thank the reviewer for the valuable question. With all due respect, we wish to clarify that NEP fundamentally differs from existing autoregressive (AR) models in two key aspects. First, it enables editing at any position by leveraging the next token's positional embedding to "probe" the input token, while AR-based models have to generate the whole image, even if the editing is just local. Second, NEP's "editing any position" regime makes **self-improvement** during image generation possible. Please kindly refer to our response to Reviewer KK48 W4 and Sec. 3.4 in the main text for test-time scaling results. These distinctions enable NEP to deliver robust, real-world performance that surpasses AR models.
>
>
> > W3. The paper lacks an analysis of the computational overhead associated with the proposed method. Important details, such as memory usage (in GB), training and inference runtimes (in seconds), should be explicitly mentioned to quantitatively assess efficiency of NEP.
>
> We thank the reviewer for raising this insightful question, as validating the inference efficiency is crucial to our contribution. We have listed these numbers below:
>
> |Methods|GPU hours (h)| Memory (GB) | Inference time (s) |
> |-|-|-|-|
> |UltraEdit|Unavailable|4.04|2.94|
> |EditAR|Unavailable|6.59|10.70|
> |NEP|1560|13.25|2.88|
>
> NEP requires higher GPU resources due to the concatenation of mask embeddings along the sequential dimension (Section 2.2), which increases sequence length and attention computational cost. Despite this, our approach achieves the fastest editing speed as we only need to predict editing region tokens rather than the whole image as AR-based models do. We have included this analysis in the revised manuscript. Thank you for your valuable input, and we welcome further suggestions to enhance our discussion.
>
>
> > W4. The overall writing and presentation are unclear in several parts, making it hard to understand the main strategy and contributions of the paper. A more detailed and intuitive explanation of the NEP framework, supported by a well-structured figure (i.e. improving Figure 2 for intuitively explaining Eq. (4)), would significantly enhance clarity.
>
> We apologize for the confusion and sincerely thank the reviewer for the valuable feedback. We have proofread our manuscript again to resolve all confusing parts that we identify. Following the suggestion, we have also improved Figure 2 for better illustration.
>
> > W5. The organization of the paper could be improved. For example, moving the related work section into Section 2 would provide a more logical flow, since the related work section is essential to fully understand the method. Also, distincting the preliminaries and the proposed method section would help readers to better understand the paper.
>
> Thank you for the feedback on the paper's organization. We agree that restructuring the paper could enhance its clarity and logical flow. Specifically, we have moved the related work section to Section 2 to provide better context for the method. We will also put "Preliminaries on LlamaGen" and "RLlamaGen: Randomized Autoregressive Text-to-Image Generation" into two subsections if the space permits.
>
> > Q1. Could the authors explore alternative strategies for NEP beyond the one described in Eq. (3)? For instance, are there different methods to represent or encode the editing regions that might yield better performance or flexibility?
>
> We thank the reviewer for their insightful question. An alternative approach to encoding the editing region, as employed in InstructPix2Pix [2], involves treating the editing region as an additional image and concatenating it with the source image along the channel dimension. For InstructPix2Pix, this is straightforward, as it only requires modifying the first CNN layer to accommodate the increased input channel size. However, for transformer-based models, this approach is less flexible. Since the image token representation is directly projected from token IDs to the transformer’s hidden size, incorporating additional channels necessitates either a new projection layer to align with the hidden size or a modification of the hidden size across all transformer blocks. In our work, we opt to integrate masks by concatenating them along the sequential dimension after tokenizing them as embeddings matching the transformer’s hidden size. This approach enhances flexibility while maintaining compatibility with the model architecture. We have elaborated on this design choice in the revised manuscript and welcome further suggestions from the reviewer to improve flexibility or performance. Thank you for your valuable feedback.
>
> [1] Mu, Jiteng, et al. Editar: Unified conditional generation with autoregressive models. CVPR 2025.
>
> [2] Brooks, Tim, et al. Instructpix2pix: Learning to follow image editing instructions. CVPR 2023.

---

> > ### Comment · Reviewer_Zgfx · 2025-08-01
> > **Thanks for the rebuttal**
> >
> > I appreciate the authors’ effort during the rebuttal process. My concerns have been well addressed, especially in the following aspects:
> >
> > **W1.** The authors compared NEP with additional baselines including EditAR, aMUSEd, Painter, and RAVG (as well as the experiments provided in response to reviewer KK48). NEP consistently outperformed these baselines, which strengthens the effectiveness of the proposed method.
> >
> > **W2.** Thank you for clarifying the difference between NEP and existing AR image editing models. The authors highlighted two main aspects; (1) NEP allows image editing from any position, and (2) NEP allows for self-improvement of image. I also appreciate the authors’ revision for test-time scaling results, which were honestly reported in the response to reviewer KK48.
> >
> > **W3.** I thank the authors for providing quantitative evaluation of computational overhead. Notably, NEP achieves the lowest inference time across baselines (UltraEdit, EditAR), which is a promising outcome.
> >
> > **W4, W5, Q1**. Thanks for giving meaningful discussion and thoughtful feedback.
> >
> > In conclusion, considering all of the above, I’ve decided to increase my score from 3 to 4.

---

### Official Review · Reviewer_TJgQ · 2025-07-07

**Clarity:** 2
**Significance:** 2
**Originality:** 2
**Rating:** 4
**Confidence:** 5

**Summary:**

Next Editing-token Prediction (NEP) is an innovative autoregressive approach to text-guided image editing that simply regenerates the tokens under user-defined editing masks, so the proposed method doesn't have to produce the whole image. This strategy fixes problems with inefficiencies and unplanned changes that are frequent in diffusion-based approaches. NEP goes through two steps of training: first, trains RLlamaGen, a randomized autoregressive text-to-image model that can generate tokens in any sequence, and then fine-tunes for localized editing using the UltraEdit dataset. NEP can do zero-shot editing, region-based control, and test-time iterative refining. The proposed method gets excellent results on MagicBrush and is competitive on EmuEdit.

**Questions:**

1. Mask Robustness: How does NEP handle noisy or misaligned editing masks? Have you tested its robustness to common segmentation imperfections?

2. Computational Efficiency: Can you detail NEP’s GPU hours, memory usage, and inference cost including the impact of iterative refinement compared to diffusion or AR baselines?

3. Generalization & Failures: Beyond UltraEdit, how does NEP perform on natural, real-world images and masks? Can you share any failure case studies?

**Ethical Concerns:**

["NO or VERY MINOR ethics concerns only"]

**Final Justification:**

The author addressed most of my concerns.

**Limitations:**

While NEP offers strong editing performance, it relies on synthetic training data and clean user-provided masks, which may limit its generalization to real-world scenarios involving noisy or imperfect inputs. The model’s robustness to such imperfections is not evaluated. Its autoregressive nature introduces error accumulation during generation, and although iterative refinement helps, it adds inference overhead.

**Quality:**

2

**Strengths And Weaknesses:**

Strengths:

1. Efficient and Targeted Editing: NEP regenerates only user-defined masked regions, avoid unnecessary full-image reconstruction, which makes NEP computationally  friendly, it also eliminates editing bias common in diffusion-based approaches.

2. State-of-the-Art Results: NEP achieves good performance on MagicBrush (L1: 0.0547 single-turn, 0.0707 multi-turn) and performs competitively on EmuEdit.

3. Well-Designed Framework: The two-stage pipeline pre-training RLlamaGen for arbitrary-order generation and fine-tuning for editing supports both zero-shot and fine-grained editing. Test-time iterative refinement further boosts quality in a scalable way.

Weaknesses:

1. Baseline Coverage: Lacks evaluation against recent AR editing methods like EditAR [1] and ControlVAR [2], which leads to gaps in comparative analysis.

2. Synthetic-Only Training Data: Fully reliant on UltraEdit-generated masks, which may not reflect real-world editing noise and could limit generalization of the NEP.

3. Engineering Over Invention: While NEP shows a robust and practical autoregressive image editing method, efficient mask-constrained editing and gradual refinement, NEP stands on the shoulders of existing techniques rather than forging a new theoretical frontier in generative modeling.

*References*

[1] Mu et al. *EditAR: Unified Conditional Generation with Autoregressive Models*, arXiv:2501.04699

[2] Li et al. *ControlVAR: Exploring Controllable Visual Autoregressive Modeling*, arXiv:2406.09750

---

> ### Author Rebuttal · Authors · 2025-07-30
>
> We appreciate the reviewer’s constructive and encouraging feedback on the novelty, efficiency, and effectiveness of NEP. To address your concerns, we have included additional comparative results to further demonstrate NEP’s empirical effectiveness (W1) and efficiency (Q2). We have also clarified its generalization ability (W2, Q3). Furthermore, we have expanded our discussion on distinctions from prior works (W3) and elaborated on limitations, particularly regarding reliance on user-specified masks (Q1). For all raised concerns, we have listed our point-to-point responses in the following.
>
> > W1. Baseline Coverage: Lacks evaluation against recent AR editing methods like EditAR [1] and ControlVAR [2], which leads to gaps in comparative analysis.
>
> Thank you for pointing out this important question. We agree that including more comparitive results with recent advances is beneficial to the solidness of our work. However, we respectfully wish to clarify the distinction between our work and ControlVAR. ControlVAR primarily focuses on image-driven editing, utilizing visual inputs such as canny edge maps, depth maps, or segmentation maps as the primary guidance. In contrast, our approach centers on text-driven editing, where textual descriptions or prompts serve as the guiding input. For a detailed discussion of the differences between these two tasks, please refer to Section 3.1.4 of the survey [1]. Due to their distinct input modalities and objectives, these methods address different problem domains and are not directly comparable.
>
> Therefore, we only comprare with EditAR. As EditAR was not accepted at the time of our initial submission, it was not included in the original manuscript. In response to your feedback, we have conducted a comparative experiment on the Magicbrush dataset for the revised version. We use EditAR’s publicly available checkpoint, inference code, and default hyperparameters for the editing task. The results are presented below:
>
>
> | Settings | Methods | L1↓ | L2↓ | CLIP-I↑ | DINO↑ |
> | --- | --- | --- | --- | --- | --- |
> | Single-turn | EditAR | 0.1028 | 0.0285 | 0.8679 | 0.8042 |
> | Single-turn | Ours | 0.0547 | 0.0163 | 0.9350 | 0.9044 |
> | Multi-turn | EditAR | 0.1341 | 0.0433 | 0.8256 | 0.7200 |
> | Multi-turn | Ours | 0.0707 | 0.0269 | 0.9107 | 0.8493 |
>
>
> Our approach demonstrates significantly improved performance. This advantage stems from NEP’s optimization, which focuses exclusively on editing and regeneration tasks. In contrast, EditAR must balance both regeneration and reconstruction, which can compromise its editing performance.
>
> We have included this comparative results and clarified this distinction in the revised manuscript, to better highlight NEP’s strengths. We appreciate the reviewer’s valuable input and welcome further suggestions to refine our analysis.
>
> Please kindly refer to our response to Reviewer KK48 W1 for more comparative experiments.
>
>
> [1] Xiong, Jing, et al. Autoregressive models in vision: A survey. arXiv preprint arXiv:2411.05902 (2024)
>
>
>
>
>
>
> > W2. Synthetic-Only Training Data: ...could limit generalization of the NEP.
>
> We sincerely thank the reviewer for their insightful question. We respectfully wish to clarify that using synthetic datasets for training editing models is a standard practice due to the high cost and complexity of acquiring real-world editing datasets. We select UltraEdit as our training dataset given its widespread use and established relevance in the field. The strong performance of our model, NEP, on real-world test sets such as Magicbrush and Emu Edit demonstrates its robust generalization capabilities. We welcome any suggestions from the reviewer regarding suitable real-world datasets that could further enhance our training pipeline.
>
> > W3. Engineering Over Invention: ... NEP stands on the shoulders of existing techniques rather than forging a new theoretical frontier in generative modeling.
>
>
> Thank you for acknowledging the strengths of our method. We respectfully wish to clarify that NEP fundamentally differs from existing autoregressive (AR) models in two key aspects. First, it enables editing at any position by leveraging the next token's positional embedding to "probe" the input token, while AR-based models have to generate the whole image, even if the editing is just local. Second, NEP's "editing any position" regime makes **self-improvement** during image generation possible. Please kindly refer to our response to Reviewer KK48 W4 and Sec. 3.4 in the main text for test-time scaling results. These distinctions enable NEP to deliver robust, real-world performance that surpasses AR models. We are grateful for the reviewer’s insights and are eager to further refine our approach while exploring new theoretical advancements in generative modeling.
>
> > Q1. Mask Robustness: How does NEP handle noisy or misaligned editing masks? Have you tested its robustness to common segmentation imperfections?
>
> Thank you for raising this important question. We agree that further expanding discussions on NEP’s dependence on masks enhances the manuscript. As noted in the limitations section, NEP relies on accurate editing region masks. In cases where user-specified masks are imperfect, we identify two primary scenarios: 1) the segmentation mask is larger than ground truth editing region, and 2) the segmentation mask is smaller. For the first scenario, NEP demonstrates robustness, achieving comparable results, as shown in Table 2 in the main manuscript, where we tested an free-form image editing without mask region. For the second scenario, our current approach is not specifically optimized. In response to your feedback, we have added a discussion of this limitation in the revised manuscript and plan to develop an editing pipeline that automatically refines user-specified masks in future work. We appreciate your valuable input and believe these revisions strengthen our contribution.
>
>
>
>
> > Q2. Computational Efficiency: Can you detail NEP’s GPU hours, memory usage, and inference cost including the impact of iterative refinement compared to diffusion or AR baselines?
>
> We thank the reviewer for raising this insightful question, as validating the inference efficiency is crucial to our contribution. Please see the following for the evaluation results:
>
> |Methods|GPU hours (h)| Memory (GB) | Inference time (s) |
> |-|-|-|-|
> |UltraEdit|Unavailable|4.04|2.94|
> |EditAR|Unavailable|6.59|10.70|
> |NEP|1560|13.25|2.88|
>
> NEP requires higher GPU resources due to the concatenation of mask embeddings along the sequential dimension (Section 2.2), which increases sequence length and attention computational cost. Despite this, our approach achieves the fastest editing speed as we only need to predict editing region tokens rather than the whole image as AR-based models do. We have included this analysis in the revised manuscript. Thank you for your valuable input, and we welcome further suggestions to enhance our discussion.
>
> Regarding iterative refinement, a direct comparison with autoregressive (AR) models is not feasible, as they lack support for test-time scaling. Although diffusion models can scale during testing, a fair and direct comparison remains challenging due to fundamental differences in their architectures and objectives. We have clarified this point in the revised manuscript. We appreciate the reviewer’s feedback and welcome further suggestions to strengthen our analysis.
>
>
>
> > Q3. Generalization & Failures: Beyond UltraEdit, how does NEP perform on natural, real-world images and masks? Can you share any failure case studies?
>
> We sincerely thank the reviewer for the valuable question. To clarify, all our evaluations were conducted on natural images from established open-source benchmarks, including MagicBrush and Emu-Edit. The qualitative comparisons presented in Tables 1 and 2 of the main manuscript demonstrate NEP’s strong generalization from synthetic to real-world datasets. Regarding failure cases, we are unable to share visualized results due to this year’s rebuttal policy. However, in response to your suggestion, we have included a detailed analysis of failure cases in the revised manuscript. We appreciate your feedback and welcome further suggestions to enhance our work.

---

### Note · Authors · 2025-08-12

We sincerely appreciate the reviewers’ time and their constructive, encouraging feedback on our paper: NEP is **innovative** and **well-designed** and achieves **good performance** on MagicBrush (Reviewer TJgQ); NEP is a **novel** framework and demonstrates **strong experimental results** (Reviewer Zgfx); this paper presents a **novel efficient** way, and it delivers controllable, fine-grained edits with **higher faithfulness** and markedly **better compute efficiency** (Reviewer 9n1p); the idea of NEP is **technically sound**, and the paper has **successfully demonstrated its capabilities** (Reviewer KK48).

We are grateful for their insightful comments and have incorporated additional discussions and experiments in response. Our main revisions are summarized as follows:
1. We added more comparative experiments with EditAR (Reviewer TJgQ, Zgfx), a recently published autoregressive image editing approach, as well as other approaches such as MaskGIT, Painter and RAVG (Reviewer KK48).
2. We validated NEP’s computational efficiency (Reviewer TJgQ, Zgfx, KK48)
3. We expanded our discussion on distinctions from prior works (Reviewer TJgQ, Zgfx) and elaborated on limitations, particularly regarding reliance on user-specified masks (Reviewer TJgQ, KK48).
4. We improved the manuscript’s clarity and organization (Reviewer Zgfx), included additional implementation details (Reviewer KK48), and elaborated on design choices (Reviewer 9n1p).


We have resolved all the concerns raised by reviewers who have participated in the discussions: "My concerns have been well addressed; I’ve decided to increase my score from 3 to 4" (Reviewer Zgfx); "The responses address my questions" (Reviewer 9n1p); "My concerns are well addressed, and I will increase my score accordingly" (Reviewer KK48). We truly appreciate the opportunity to improve this work based on the reviewers' valuable insights.

---

### Decision · Program_Chairs · 2025-09-17

**Decision:**

Accept (poster)

**Comment:**

This work introduces NEP (next editing token prediction), where only to-be-edited regions are regenerated, avoiding unintended modifications to other regions. To enable any region editing for autoregressive models, the authors pretrain a randomized autoregressive model for text-to-image. Promising results are demonstrated on MagicBrush and Emu Edit.

Initially, the reviewers raised several concerns, which are outlined below:

* Reviewer TJgQ: Lack of recent baselines for comparisons, reliance on synthetic training data, limited novelty.

* Reviewer Zgfx: Lack of other baselines, limited novelty, lack of discussion on computational overhead, writing issues.

* Reviewer 9n1p: Explanation on performance drop in Tab. 3, output orders.

* Reviewer KK48: Lack of other baselines, latency comparison, weak performance on free-form editing and test-time scaling, reliance on user-provided masks, lack of implementation details.

The rebuttal and subsequent author-reviewer discussions effectively addressed most of the reviewers' concerns. After carefully considering the reviews, rebuttal, and discussion, the AC concurs with the reviewers’ assessment and thus recommends acceptance of the paper.

Additionally, the AC kindly notes that in addition to [52], the work RandAR [A] also enables any-region editing by training autoregressive models with random permutation. A proper discussion and citation of the related work would strengthen the revision. Finally, the authors are encouraged to incorporate the rebuttal experiments into the manuscript and address the reviewers’ feedback in the final revision.

[A] RandAR: Decoder-only Autoregressive Visual Generation in Random Orders. CVPR 2025.